# Spatial Delimitation of Genetic Diversity of Native Maize and Its Relationship with Ethnic Groups in Mexico

Alberto Santillán-Fernández [1,2], Yolanda Salinas-Moreno [3], José René Valdez-Lazalde [4], Jaime Bautista-Ortega [5] and Santiago Pereira-Lorenzo [6,*]

1 Catedrático-Conacyt, Colegio de Postgraduados Campus Campeche, 24450 Champotón, Campeche, Mexico; santillan.alberto@colpos.mx
2 International Doctorate Program of Agricultural and Environmental Sciences, Escuela Politécnica Superior, Universidad de Santiago de Compostela, Galicia, 27002 Lugo, Spain
3 Department of Genetic, Instituto Nacional de Investigaciones Forestales, Agrícolas y Pecuarias, Campus Altos de Jalisco, 47600 Tepatitlán de Morelos, Jalisco, Mexico; salinas.yolanda@inifap.gob.mx
4 Department of Forestry, Colegio de Postgraduados Campus Montecillo, 56230 Texcoco, Mexico; valdez@colpos.mx
5 Department of Agricultural Sciences, Colegio de Postgraduados Campus Campeche, 24450 Champotón, Campeche, Mexico; jbautista@colpos.mx
6 Department of Plant Production and Engineering Projects, Escuela Politécnica Superior, Universidad de Santiago de Compostela, Galicia, 27002 Lugo, Spain
* Correspondence: santiago.pereira.lorenzo@usc.es; Tel.: +34-629-83-71-26

**Abstract:** Mexico, as a center of origin of maize, presents a high diversity of maize races. With the objective of spatially demarcating regions with high concentration of intraspecific diversity in Mexico, as a fundamental measure for the in situ conservation of their agrobiodiversity, Geographic Information Systems (GIS) tools have been used to generate diversity and richness indexes for 64 maize races cultivated in Mexico, using indexes to demarcate relationships to environmental factors such as temperature, precipitation, and altitude; the presence of indigenous groups; and the type of maize used. These relations allowed defining seven environmental units spatially, with characteristic maize races in each of them, which constitute priority areas for in situ conservation. In addition, a close relationship was found between the diversity of maize races and of ethnic groups, from the center to the south of the country, associated with the differences in maize use in different ethnic groups. This geographical demarcation of races and uses of maize will favor food security through specific measures of in situ conservation, as well as an increase in added value of maize by-products based on specific maize races conserved by local ethnic groups.

**Keywords:** *Zea mays* L.; diversity of crops; maize races; intraspecific diversity

## 1. Introduction

Mexico is considered the center of origin of maize (*Zea mays* L.) and is home to a large diversity of maize races, each one adapted to different environmental conditions and agricultural systems [1,2]. These races are cultivated in a broad range of altitudes and precipitation regimes (from sea level up to around 3000 masl, of less than 400 mm to more than 3000 mm of rainfall per year). As a result, the races vary in plant size and tolerance to heat, cold, and water needs [3].

Around 64 maize races are cultivated in Mexico [4], representing 60% of maize diversity in the world [1]. The importance of maize for the Mexican population has been thoroughly described, from several points of view, as center of origin [5–7], center of domestication [8,9], basic food [10], and even as a factor that has allowed the development of ethnic groups [3,11].

The analysis of biodiversity in the centers of origin is crucial for the in situ conservation of species [12–15]. However, studies at the national level to identify priority areas for in situ

conservation of genetic maize resources in Mexico are scarce [16,17]. Most studies have centered on ex situ conservation through the creation of germplasm banks and have been limited to describing the geographic distribution of the races [4,12]. Studies about the interaction between environmental, sociocultural, and economic factors that impact maize agricultural production systems most are also limited [3,18–20].

Identification of priority areas for in situ conservation can help to plan agricultural activities better and decrease the vulnerability of production when facing environmental threats [3,21,22]. It also contributes to satisfying specific cultural or social needs and even to stopping the adoption of new technologies such as genetically modified organisms that compromise the diversity of the species [23–25].

The authors of [26] found evidence of loss of maize varieties in Mexico, which is why they suggested that the reduction of varietal diversity could make it difficult for farmers to select races adapted to current climate variation, reducing the field yields. These losses in yield would lead, in turn, to farmers abandoning the sector and, therefore, to a greater loss of maize diversity in Mexico.

The authors of [27,28] suggest that, in order to preserve species diversity, the spatial analysis using Geographic Information Systems (GIS) can be a good tool when different types of spatial and temporal information are connected, as shown in studies about potato [29], papaya [30], manioc [31], and, more recently, maize [3,7,18–20], in particular, when richness and diversity indexes are incorporated that allow demarcating priority areas for the conservation of maize agrodiversity, analyzed from a territorial approach [3]. Due to the simplicity of their calculation and the reliability of their results, the Shannon, Simpson, and Margalef diversity indexes are commonly used in studies on conservation and use of plant genetics resources [28].

Recently, the authors of [7,18,19] analyzed the geographical variation of maize diversity in Mexico using indicators of richness and diversity. However, in their studies they did not analyze intraspecific diversity or the relationship of ethnic groups with the territorial diversity of races and specific maize uses.

To address this problem, in this study we set out the objective of demarcating regions with a high concentration of intraspecific diversity of maize in Mexico, through GIS tools that relate environmental factors, such as temperature, precipitation, and altitude, with the presence of ethnic groups and the type of maize used. The initial hypothesis was that the diversity of maize races is associated with the territorial diversity of ethnic groups, contributing to the domestication process, with specific uses per race. This geographic delimitation of maize races and uses would favor food security through specific measures for conservation, as well as an increase in the added value of maize by-products based on specific maize races conserved by local ethnic groups.

## 2. Materials and Methods

### 2.1. Databases

In this study, 18,812 georeferenced records of 64 cultivated maize races in Mexico were included, and 1570 of its wild relative, teosinte [32], were integrated in a database available on the website of the National Commission for the Knowledge and Use of Biodiversity (CONABIO) [4]. The 20,382 records have an identifier that groups them into maize and teosinte races (Figure 1), year of collection (1927 to 2010), maize uses (grain and plant), and geographic coordinates (longitude, latitude).

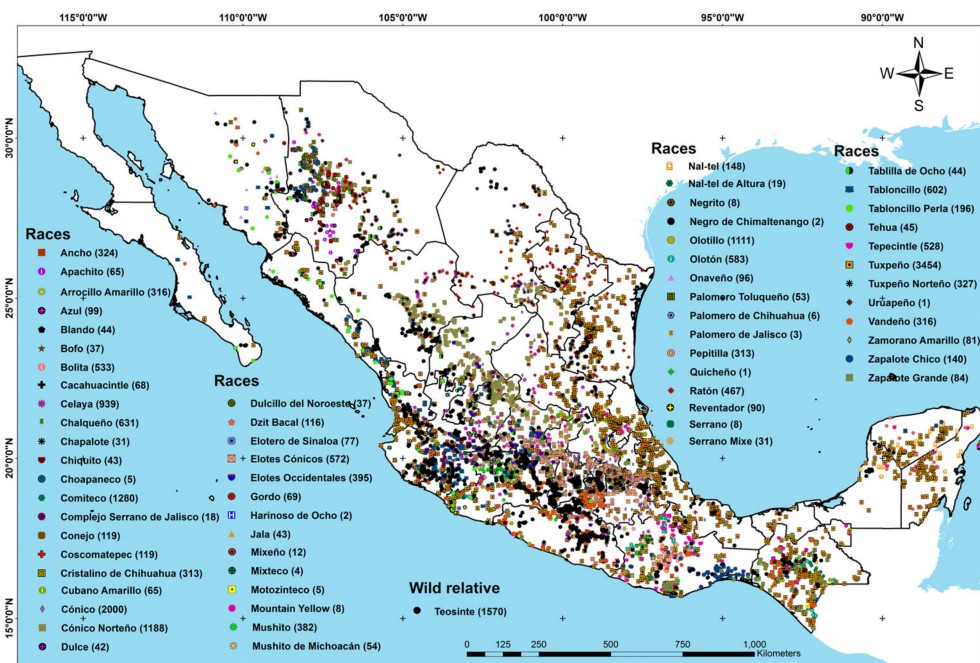

**Figure 1.** Spatial distribution of maize races in Mexico and their wild relative teosinte. The number of records per race is shown in parentheses.

### 2.2. Spatial Delimitation of the Regions of Genetic Richness per Maize Race

The methodology used to determine genetic richness zones was based on the spatial analysis of the geographic distribution of the collections of 64 maize races. For this purpose, the collections per race were grouped and their genetic richness zone was modelled in the software DivaGis v7.5 [33] with the module Analysis/Point to Grid/Richness. In the case of the races with low representativeness (less than 30 collections), buffer zones of 5-km radius were generated for each collection, because it was considered that there is high probability that the genetic material collected is distributed in this area [34].

The richness zone for the races with more than 30 collections was calculated race by race and consisted in dividing the country's map into grids of five min (0.083 sexagesimal degrees) with a vicinity of one sexagesimal degree that is equivalent to an area of 9 km by 9 km, a size recommended for national studies [35]. DivaGis makes a count of the number of collections in each grid and groups the grids into five class intervals by the number of observations present. For purposes of this study, four class intervals were taken with higher number of observations.

According to the methodology described for the determination of centers of origin and centers of genetic diversity of maize in Mexico [36], the regions generated by the buffer zones for the collections of races with less than 30 observations and the richness zones for the races with more than 30 observations were intersected with the records of wild relatives (teosinte) and those regions without presence of wild relatives were distinguished. The resulting regions were called genetic richness regions.

### 2.3. Spatial Delimitation of Intraspecific Maize Diversity

To determine the spatial intraspecific diversity of the 64 maize races grown in Mexico, the geographic distribution of their collections was used and specific richness and structure indexes were modeled in the software DivaGis v7.5 [33] with the Analysis/Point to Grid/Diversity module. To calculate the indexes, the country's map was divided into grids of five min (0.083 sexagesimal degrees) with a vicinity of one sexagesimal degree that is equivalent to an area of 9 km by 9 km [35].

The specific richness indexes that were calculated were those by Margalef and Menhinick, which measured the number of races present per grid (9 km by 9 km) without taking

into consideration the proportional abundance of their collections [37]. For the structure indexes, those by Shannon, Brillouin, and Simpson were calculated, which, in contrast with the specific richness indexes, did consider the proportion of the number of collections per maize race inside the grid [37].

According to the index used, DivaGis counts the number of races in each grid and groups the grids into five class intervals in ascendant order by the number of races present [33]. To simplify the interpretation of the results, the five class intervals per index were reclassified as very low diversity, which corresponded to the class interval (1), low (class interval (2)), medium (3), high (4), and very high (5).

For the purposes of this study, the common areas with medium, high, and very high diversity of the specific richness indexes (Margalef and Menhinick) and structure indexes (Shannon, Brillouin, and Simpson) were taken into account, which were demarcated through a process of map intersection in DivaGis v7.5 [33]. The resulting polygons were called regions with intraspecific diversity of maize.

### 2.4. Spatial Delimitation of the Maize Diversity Center

Reference [36] defines diversity centers as those regions where populations of wild relatives coexist with populations of races or varieties of the cultivated species. In this study, the maize diversity center was defined spatially based on the union of the georeferenced regions of: (1) genetic richness, (2) intraspecific diversity, and (3) centers of origin and domestication of maize described by Kato for Mexico [38].

### 2.5. Characterization of the Maize Diversity Center by Climate Factors

Considering that the maize diversity center identified covers different regions of Mexico, each with different climate patterns that give special characteristics to the maize races cultivated in these regions [1], a regionalization was made to differentiate the climatic conditions inside the diversity center. For this purpose, the geographic layer of the diversity center was combined with the geographic layers of altitude (masl), mean annual precipitation (mm), mean annual temperature (°C), and temperature regime (with categories from very dry to very humid) available in CONABIO's geoportal [39]. The geographic layers were combined with the intersect tool of the GvSIG v2.3.1 software [40], according to the Methodology of Environmental Units by Gómez-Orea [41].

### 2.6. Relationship of Environmental Units with Ethnic Groups and Maize Uses

Once the environmental units were characterized by climatic factors, the principal races of maize cultivated in each of them were identified. Therefore, the number of collections per maize race in each environmental unit was counted and the percentage represented by the total per race was obtained. The races whose percentage of collections in the environmental unit was higher compared to the other environmental units were considered as principal races. For those cases where a race presented higher percentages than 10% in two or more environmental units, they were considered as a secondary race.

Once the principal races per environmental unit were determined, the ethnic groups and maize uses (grain and plant) in each environmental unit were territorially associated. The spatial distribution of ethnic groups in Mexico was obtained from the National Commission for the Development of Indigenous Peoples [42]. For the case of maize uses, 11 uses were defined for the grain and four for the plant, based on the information available in the database by CONABIO [4,10] (Table S1).

Finally, to determine the relationship that ethnic groups in Mexico have had in the past with the process of maize domestication, the number of ethnic groups in each environmental unit was associated with the corresponding number of collections, number of races, number of principal races, and number of maize uses (grain and plant). For these, Pearson's coefficient of correlation was calculated with the statistical software R [43].

In the case of the number of maize uses (grain and plant) in each environmental unit, an index of uses for principal races was heuristically constructed, with mathematical limits

in the $[0, 1]$ interval. Theoretically, the value 0 implies that out of the 15 probable uses according to CONABIO [4,10], none of the principal races in the environmental unit is used. On the contrary, the value 1 can only be observed if the environmental unit contains the total of principal races (64) and each one of them is used in the 15 possible uses.

For the creation of the index, it was reclassified as 1 if the principal race in the environmental unit presented the corresponding use and 0 in the contrary case. The mathematical expression was the following:

$$IU_i = \frac{\sum_{j=1}^{n_i} \sum_{k=1}^{15} U_{jk}}{(64)(15)} \qquad i = 1, \ldots, 7; j = 1, \ldots, n_i; k = 1, \ldots, 15$$

where

$IU_i$ = Index of uses for the environmental unit i,

$U_{jk}$ = Value given for the k use of the principal race j in the environmental unit i, and

$n_i$ = Number of principal races in the environmental unit i.

## 3. Results

### 3.1. Spatial Delimitation of the Genetic Richness Regions per Maize Race

A database analysis with 18,812 collections of the 64 maize races cultivated in Mexico allowed identifying 49 races with more than 30 records and 15 more with fewer than 30 records. Based on the methodology suggested in this study, the richness region for each race with more than 30 collections was demarcated as genetic richness region (Figure S1) and for those with fewer than 30 collections, a buffer zone of 5-km radius was generated for each collection (Figure S2).

For the sum of the regions generated by the 64 races cultivated in Mexico, the presence of the wild relative teosinte was verified, ruling out those regions without presence of teosinte (Figure S3). The resulting regions were called genetic richness regions, whose geographic distribution covered wide zones in Mexico, with greater territorial coverage in the center-south of the country (Figure 2).

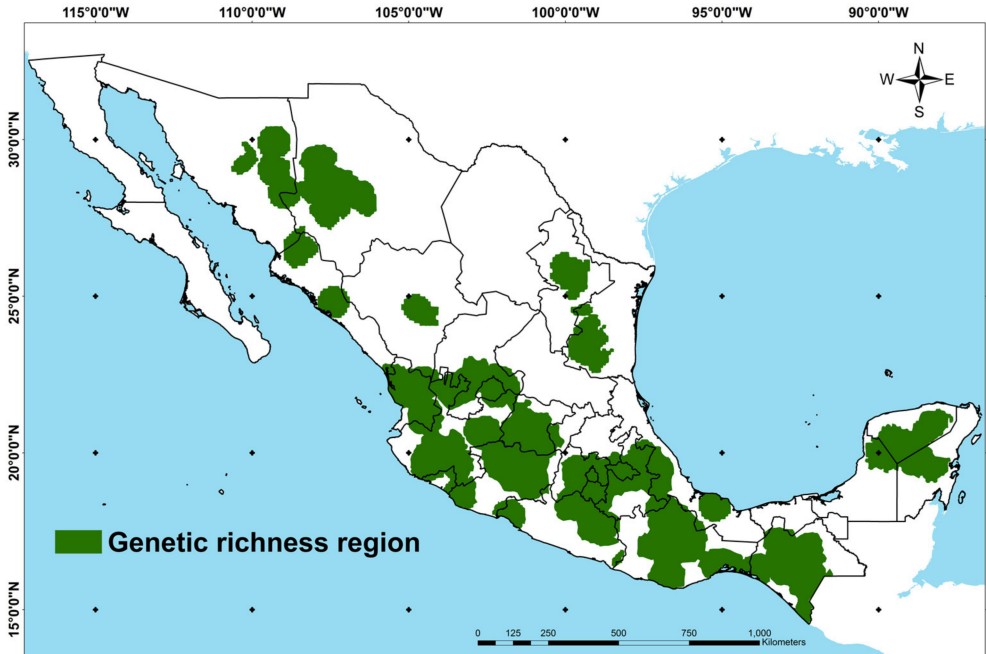

**Figure 2.** Genetic richness region for the 64 maize races cultivated in Mexico.

### 3.2. Spatial Delimitation of Intraspecific Maize Diversity

When calculating the specific richness indexes (Margalef and Menhinick) and structure indexes (Shannon, Brillouin, and Simpson) of the geographic distribution of 64 maize races cultivated in Mexico, it was found that the structure indexes demarcated broader regions of intraspecific diversity in the categories defined from high to very high. In addition, in contrast with the specific richness indexes, the structure indexes did demarcate regions of medium to high intraspecific diversity in the Yucatan Peninsula (Figure S4).

In the methodological proposal of this study, it was decided to consider only the common areas with medium to very high intraspecific diversity both of the specific richness and the structure indexes, since, according to [37,44], when considering only one diversity index there is the risk of under- or overestimating the diversity. Figure 3 shows that the common areas with medium to very high intraspecific diversity of the indexes calculated were concentrated in the center to north of Mexico.

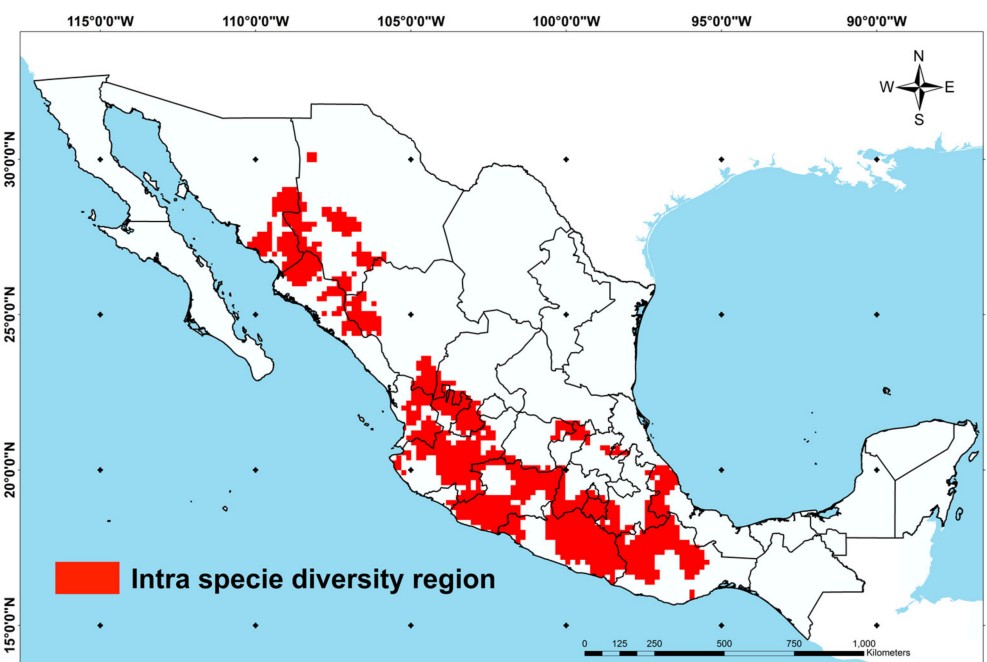

**Figure 3.** Spatial distribution of the common areas with medium to very high intraspecific diversity for the 64 races of maize cultivated in Mexico, from specific richness indexes (Margalef and Menhinick) and structure indexes (Shannon, Brillouin, and Simpson).

### 3.3. Spatial Delimitation of the Maize Diversity Center

The spatial delimitation of the maize diversity center in Mexico was defined from the union of the maize genetic richness region, the maize intraspecific diversity region, and the centers of origin and domestication of maize established for Mexico (Figure S5) [38]. Figure 4 shows that the union of the three regions allowed complementing areas, which helped to include races that may have not been contemplated by one of the three regions that make up the maize diversity center.

For example, the intraspecific diversity region did not contemplate areas in the southeast or northeast of Mexico, which were included in the genetic richness region of maize. The races Dzit Bacal and Nal-tel are grown in the southeast of Mexico, which were not cultivated in any other region of the country. The same happens for the Tuxpeño Norteño race, which is grown mostly in the northeast of Mexico [1].

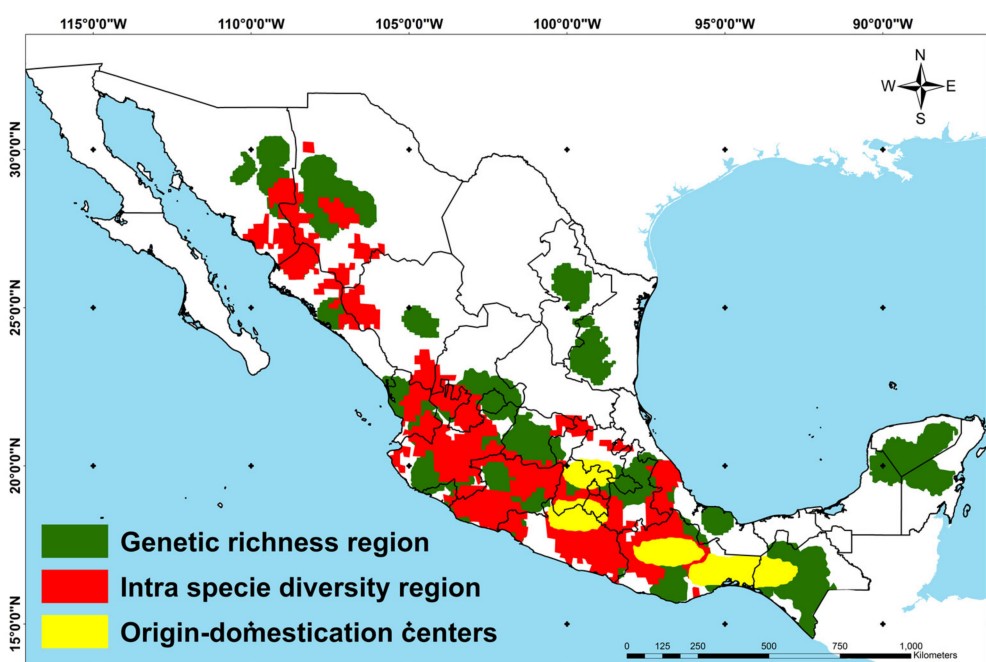

**Figure 4.** Spatial delimitation of the maize diversity center in Mexico obtained from the union of the maize genetic richness region, the maize intraspecific diversity region, and the centers of origin and domestication of maize established for Mexico.

### 3.4. Characterization of the Maize Diversity Center by Climatic Factors

The characterization by environmental units of the diversity center allowed differentiating into seven environmental units with different climatic thresholds: North Pacific, Sierra Tarahumara, Northeast, Center, South Pacific, Southeast, and Yucatan Peninsula. As shown in Figure 5, the environmental units in the north of the country were characterized by dry climates, those in the center by warm climates, and those in the south by humid climates, which confers unique characteristics to the races that are cultivated in each environmental unit [1].

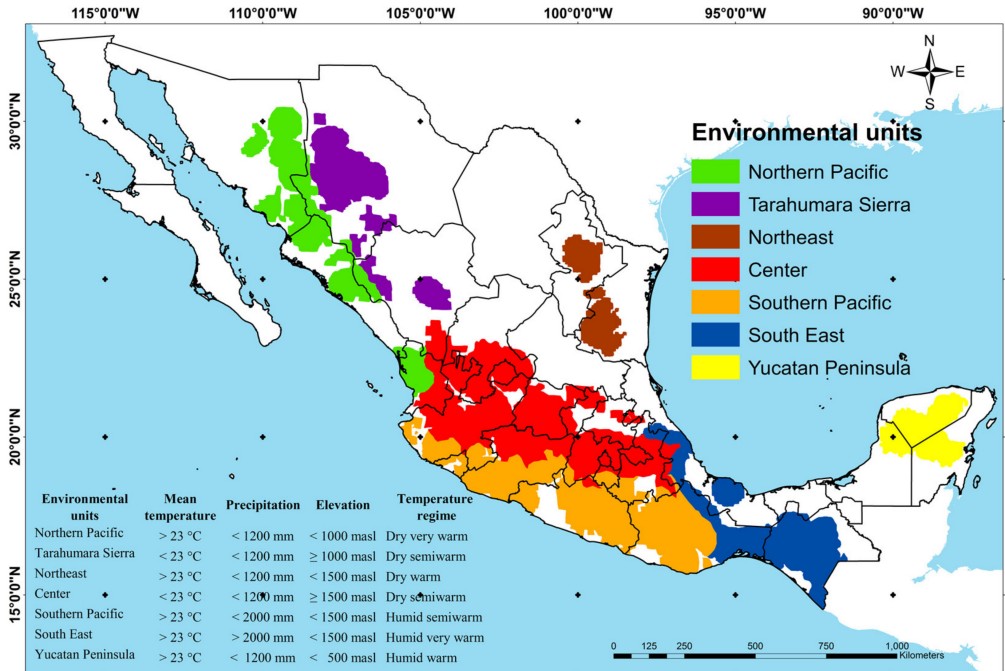

**Figure 5.** Spatial delimitation of the seven environmental units that were differentiated when characterizing the maize diversity center in Mexico by climatic factors.

### 3.5. Relationship of Environmental Units with Ethnic Groups and Maize Uses

When associating the number of collections from the total that were analyzed in this study (18,812), the number of races cultivated in Mexico (64), the number of principal races, ethnic groups, and indexes of maize grain and plant uses inside each environmental unit, it was found that 84.65% (15,925) of the total collections were included in one of the seven environmental units (Table 1). In addition, the 64 races cultivated in Mexico were grouped exclusively in the seven environmental units with specific uses of grain and plant, and ethnic groups associated territorially to the geographic distribution of the maize races. For the case of the races with broad distribution such as Ratón and Tuxpeño, it was possible to differentiate them as secondary races in one or more environmental units (Table S2).

**Table 1.** Number of collections, number of races, number of principal races, ethnic groups, and index of maize grain and plant uses inside each environmental unit.

| Environmental Unit | Collections | | Races | | Principal Races | | Ethnic Groups | | Index of Maize Grain and Plant Uses (IU) | | |
|---|---|---|---|---|---|---|---|---|---|---|---|
| | Number | % Total | Number | % Total | Number | % Total | Number | % Total | $\sum_{j=1}^{n_i} \sum_{k=1}^{15} U_{jk}$ | $(64)(15)$ | $IU_i$ |
| North Pacific | 516 | 2.74 | 25 | 39.06 | 7 | 10.94 | 7 | 11.11 | 45 | 960 | 0.05 |
| Tarahumara Sierras | 993 | 5.28 | 25 | 39.06 | 5 | 7.81 | 3 | 4.76 | 35 | 960 | 0.04 |
| Northeast | 379 | 2.01 | 8 | 12.50 | 2 | 3.13 | 0 | 0.00 | 16 | 960 | 0.02 |
| Center | 6010 | 31.95 | 49 | 76.56 | 23 | 35.94 | 18 | 28.57 | 162 | 960 | 0.17 |
| South Pacific | 3340 | 17.75 | 44 | 68.75 | 14 | 21.88 | 17 | 26.98 | 96 | 960 | 0.10 |
| South East | 4196 | 22.30 | 33 | 51.56 | 11 | 17.19 | 24 | 38.10 | 82 | 960 | 0.09 |
| Yucatán Peninsula | 491 | 2.61 | 8 | 12.50 | 2 | 3.13 | 5 | 7.94 | 17 | 960 | 0.02 |
| Sum | 15,925 | 84.65 | | | 64 | 100.00 | | | | | |
| Total | 18,812 | 100.00 | 64 | 100.00 | 64 | 100.00 | 63 | 100.00 | | | |

Table 1 shows the Center, South Pacific, and Southeast environmental units that are located in warm and semi-warm regions with annual precipitations over 1200 mm and average altitude of 1500 masl, concentrating the highest number of collections, number of races (total number of races without considering the proportion of collections in the environmental unit), and number of principal races (number of races with the highest proportion of collections in the environmental unit). In these environmental units, the highest indexes of use (grain and plant) were also obtained, and they are where territorially there is higher presence of ethnic groups (Figure S6).

Finally, the results from Pearson's correlation analysis showed that in the environmental units where there is a higher number of ethnic groups, the number of maize races and their uses (grain and plant) tend to be higher (Figure 6).

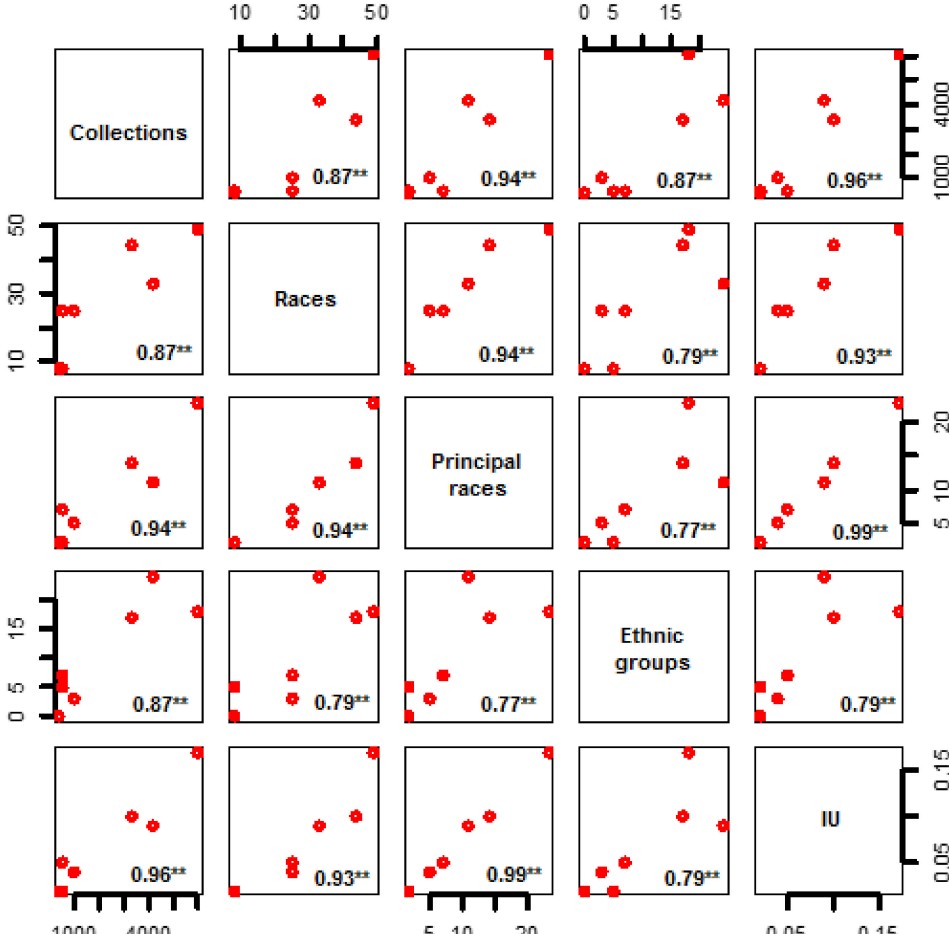

**Figure 6.** Scatterplot matrix of Pearson's correlation coefficient that associates the presence of ethnic groups in the environmental units with the number of collections, number of races, number of principal races, and index of maize uses (grain and plant) (n = 7). ** $p < 0.05$.

## 4. Discussion

### 4.1. Spatial Delimitation of the Genetic Richness Regions per Maize Race

The use of a database with the geographic distribution of 18,812 collections of the 64 maize races cultivated in Mexico allowed identifying regions of genetic richness and intraspecific diversity of maize at the national level. However, differences must be considered in the racial classification. For example, the authors of [45] reported a database for Oaxaca where they include races such as Negro de Tierra Fría, Negro Mixteco, Olotón Imbricado, and Serrano de Oaxaca. These races were not identified in the database used in this study [4].

In addition to the differences in racial classification, in the database used, 15 out of the 64 races analyzed had fewer than 30 collections, and the calculation of indexes of intraspecific diversity with GIS increases its reliability as the number of georeferenced records is higher than 30 [44]. Considering this, two specific richness indexes were calculated (which did not consider the proportional abundance of the collections) and three structure indexes (which did consider the proportional abundance of the collections), and the areas in common of the five indexes with medium to very high indexes were demarcated, which, according to [37], increase the reliability in estimating the intraspecific diversity.

In contrast with studies, such as those by the authors of [7], who defined a maize diversity center in Mexico based on the union of genetic richness regions of 54 maize races, and [18], who found 11 biogeographical regions for maize in Mexico when considering 47 maize races, our study analyzed 64 maize races and defined the diversity center as

the sum of the georeferenced regions of genetic richness, intraspecific diversity, and centers of origin and domestication of maize described by Kato [38]. This allowed demarcating a spatially larger diversity center, where the diversity of a higher number of maize races was considered.

With the methodology proposed, the diversity center was characterized into environmental units with different climatic conditions, and in each environmental unit the principal races cultivated in it were associated with maize uses (grain and plant) and ethnic groups. Therefore, in contrast with studies, such as those by the authors of [1,46], which analyzed the spatial relationships of environmental factors (altitude, climate, slope, and soil) with the spatial distribution of maize races, our study also relates these aspects with the presence of ethnic groups linked territorially to the maize crop, which is why this fact can help to redirect the in situ conservation policies in a focal and localized way.

*4.2. Environmental Units*

Because of the patterns of temperature, precipitation, and altitude, seven environmental units were differentiated with a different moisture regime. The authors of [46] found that the altitude, related to the humidity, is the factor that agrees most with the spatial distribution of maize races in Mexico.

The authors of [18], based on the racial composition of maize, defined 11 biogeographical regions in Mexico, which contain 73% of the diversity in races described by the authors of [4]. In contrast with the work by the authors of [18], our study contemplated 100% of the races described by the authors of [4] and the characterization by environmental units allowed defining climatic conditions in each one of the seven environmental units which, according to the authors of [1], condition the phenology, cultivation, and agronomic management of maize.

The authors of [19], when using the hierarchical clustering technique on the racial composition of maize, found that the highest richness and diversity of maize races in Mexico is found in the center to south of the country; this finding agrees with our results. However, the spatial resolution that they [19] used was 50 km, and in our case it was 9 km, which, according to [35], allows higher accuracy in territory delimitation.

The concentration of a higher number of maize races from the center to the south of the country is an aspect that has been widely documented [18,19,38]. However, in contrast with these studies, our study was able to establish that a higher presence of ethnic groups is associated with a higher diversity of maize races and uses. Despite the fact the sample size used for the calculation of this result is considered statistically small, according to [47], even in small samples (n < 25), the Pearson correlation coefficient allows us to know if there is relationship between the variables and what is the direction of this relationship. However, the estimate about how strong this relationship is may be biased. In light of these results, the development of future studies is suggested, to further confirm this relationship.

In this regard, the authors of [46] found that the presence of ethnic groups conditions the spatial distribution of maize races, since they are in charge of conserving, adapting, and cultivating local races for specific purposes. The authors of [11,20,48] also described a strong correlation between ethnic groups and maize diversity. However, compared to previous studies, ours describes the variety of uses given to maize grain and plant.

The whole maize plant is used in Mexico, from the grain in the preparation of foods and snacks, to the leaves, stalk, and roots as fodder, fuel, and organic fertilizer [10]. However, for Mexican people there is a basic process for the diet, nixtamalización, which gives rise to the maize dough that is used to prepare tortillas, atoles, and tamales wrapped in totomoxtle, which is the leaf that covers the cob [49].

In our study, specific uses of maize per race and ethnic group were established. Although the highest values of the index of uses were obtained in the environmental units of the center to the south of the country, environmental units were differentiated such as Sierra Tarahumara and Yucatan Peninsula where there are specific races linked ancestrally to the development of ethnic groups that are found there.

For example, the Apachito race, linked to the development of the Tarahumara ethnic group in the sierras of Chihuahua [19], is used to make tejuino, an alcoholic beverage characteristic of the north of Mexico that substitutes pulque, which is made out of maguey (*Agave salmiana*) in the center-south of the country [50]. The races Dzit Bacal and Nal-tel, which are the base of the diet of Maya people in the Yucatan Peninsula [10], are used to prepare pozol, which is a refreshing beverage that mixes cacao (*Theobroma cacao* L.) with maize [51].

## 5. Conclusions

The use of GIS tools for the territorial delimitation of genetic richness and intraspecific diversity regions of maize, and their association with ethnic groups and the uses of maize, allowed differentiating seven environmental units, in each of which the maize races with greatest presence were identified. The 64 maize races cultivated in Mexico were grouped exclusively into each of the seven environmental units: North Pacific (7 principal races), Sierra Tarahumara (5), Northeast (2), Center (23), South Pacific (14), Southeast (11), and Yucatan Peninsula (2). The greatest diversity of maize uses and races was found from the center to the south of the country, with a strong association with the ethnic groups in each of the regions. These results can help in decision making for the in situ conservation of maize races in Mexico, when identifying territories with unique characteristics linked to maize races and specific ethnic groups with traditional know-how and, therefore, ensuring food security and promoting an increase of the added value from maize by-products based on specific maize races conserved by local ethnic groups.

**Supplementary Materials:** The following are available online at https://www.mdpi.com/article/10.3390/agronomy11040672/s1, Figure S1: Delimitation of richness regions for races with more than 30 collections. Example: Cristalino de Chihuahua race. (a) Geographic distribution of collections. (b) Genetic richness region obtained through DivaGis. (c) Genetic richness region that considers the four class intervals with highest concentration of collections (31 to 59, 60 to 89, 90 to 118, and 119 to 148). Figure S2: Spatial delimitation of the buffer zones for each collection of the races with fewer than 30 records. (a) Geographic distribution of the collections of races with fewer than 30 records. (b) Spatial representation of a buffer zone of 5-km radius. (c) Spatial representation of the buffer zones of 5-km radius for each collection of races with less fewer 30 records. Figure S3: Spatial representation of the teosinte collections with the genetic richness regions of the 49 races with more than 30 collections and buffer zones of 5-km radius for the collections of the 15 races with fewer than 30 records. Figure S4: Spatial delimitation of specific richness indexes (A: Margalef and Menhinick) and structure indexes (B: Shannon, Brillouin, and Simpson) for the geographic distribution of the 64 maize races cultivated in Mexico. Figure S5. Spatial delimitation of the centers of origin and domestication of maize estab-lished by Kato in the book Origen y diversificación del maíz: una revisión analítica [38]. Figure S6: Spatial distribution of environmental units and ethnic groups in Mexico. Table S1: Description of the principal uses of the grain and the plant of 64 native maize races cultivated in Mexico. Table S2: Relationship of the principal maize races cultivated in each environmental unit with specific uses of grain and plant and the ethnic groups associated territorially to the geographic distribution of maize races.

**Author Contributions:** Conceptualization, information analysis, and writing of original draft, A.S.-F.; data review and monitoring results, Y.S.-M.; information analysis and writing of final manuscript, J.R.V.-L.; writing, revising, and editing of the final manuscript, J.B.-O.; writing, revising, and editing of the final manuscript, S.P.-L. All authors have read and agreed to the published version of the manuscript.

**Funding:** This research received no external funding.

**Institutional Review Board Statement:** Not applicable.

**Informed Consent Statement:** Not applicable.

**Data Availability Statement:** Availability of data. The data are available with the first author, at reasonable request.

**Acknowledgments:** This study is part of the doctoral thesis of the first author, in the International Doctorate Program of Agricultural and Environmental Sciences of the Universidad de Santiago de Compostela, Spain, and to Project number 364. Sustainable productive reconversion for the development of rural producers in Campeche was assigned to the first author by the Consejo Nacional de Ciencia y Tecnologia (CONACyT). To the anonymous reviewers of the article, for their comments, which helped to enrich the research.

**Conflicts of Interest:** The authors declare no conflict of interest.

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
