# Peer review of "Spatial Delimitation of Genetic Diversity of Native Maize and Its Relationship with Ethnic Groups in Mexico"

_agronomy, doi:10.3390/agronomy11040672_

Round 1

Reviewer 1 Report

The paper is simple, yet elegant. There are several grammar and typing flaws and some information is missing. I will suggest it for publication after addressing several of my minor comments stated bellow.

  • In introduction, at least some discussion on selection and performance of specific indices is needed

M&M section is generally very vague

  • More details should be given on sampling in the dataset. How were the samples collected, how was it controlled for bias in sampling, why only 64 cultivars? Is there data on more? Please explain how this affects bias in sampling?
  • Teosinte or teocinte? However, it seems underrepresented. Please comment
  • More details should be given to descriptions and calculation of diversity indices
  • Figure 1 resolution is poor

Figure 4 shows colours beyond the ones in legend

Please state N in Table 2 for correlation analysis. According to the p values and strengths of correlations, correlations are overestimated. This should be stated in the text and conclusions based on these data should be treated with caution

Author Response

Dear Reviewer

The Authors reply to:

Comments and Suggestions for Authors

The paper is simple, yet elegant. There are several grammar and typing flaws and some information is missing. I will suggest it for publication after addressing several of my minor comments stated bellow.

  • In introduction, at least some discussion on selection and performance of specific indices is needed

Authors reply: Due to the simplicity of their calculation and the reliability of their results, the Shannon, Simpson and Margalef diversity indexes are common used in studies on conservation and use of plant genetics resources [28]. We include this precision in the text.

M&M section is generally very vague

  • More details should be given on sampling in the dataset. How were the samples collected, how was it controlled for bias in sampling, why only 64 cultivars? Is there data on more? Please explain how this affects bias in sampling?.

Authors reply: Dear reviewer. We did not realice any sampling for obtaining the accesiones.  For the analysis, we employed the oficial information about these accesiones that is available in the web page of the Nationanl Comission for Knowledge and use of the biodiversity (CONABIO), reference 4.  

  • Teosinte or teocinte? However, it seems underrepresented. Please comment

Authors reply: The word teocinte was changed by teosinte (no italics), which is the right term to refers to the wild ancestors of maize. The legends in fig. 1 and fig. S3, were also changed. We include this precision in the text.

  • More details should be given to descriptions and calculation of diversity indices.

Authors reply: Dear reviewer on the lines 119-140 of the section 2.3 “Spatial delimitation of intraspecific maize diversity” the types of diversity indices used for this study are detailed and their calculation in the Diva Gis software is specified.

  • Figure 1 resolution is poor

Authors reply: Figure 1 was corrected.  The legends for maize races were put in bold, for upgrading visualization. The resolution of this figure was improved (300 dpi). We include this precision in the text.

  • Figure 4 shows colours beyond the ones in legend

Authors reply: The Figure 4 was corrected.  The shadow in the figure was eliminated. We include this precision in the text.

  • Please state N in Table 2 for correlation analysis. According to the p values and strengths of correlations, correlations are overestimated. This should be stated in the text and conclusions based on these data should be treated with caution

Authors reply: Dear Reviewer, thanks for this suggestion. To ensure that Pearson's correlation coefficient is appropriate, a Scatterplot matrix was created, The Table 2 was changed by Figure 6. We include this precision in the text.

Authors reply: In the legend of Figure 6 the N information used for calculating the Pearson correlation coefficient (n=7) was added. We include this precision in the text.

Authors reply: Figure 6. Scatterplot matrix of Pearson’s correlation coefficient that associates the presence of ethnic groups in the environmental units with the number of collections, number of races, number of principal races, and index of maize uses (grain and plant) (n=7). We include this precision in the text.

Best regards

The Authors

Reviewer 2 Report

I found this paper very difficult to read.

After several readings the methods were still unclear. The authors refer often to the use of the DivGis program to explain the methods which were not well explained  for a non DivGis user which I think should be done for a journal in Agronomy.

The methods should be explained without the need to refer to a program. 

In section 2.6 , line 179 the authors state "For  the creation of the index, it was re-categorized ... "  . Replace 'it' with a more descriptive term.

The legend colors in Figure 4 need to be corrected.

I am very concerned about the use of Pearson correlation coefficient to measure associations between the presence of ethnic groups with the number of collections. 

Pearson's correlation coefficient assumes a simple linear relations between the various measures.  I would like the authors to include a scatterplot matrix  to assure the reader that this statistic is appropriate.

I would recommend the scatterplotMatrix function from the 'car' package available in CRAN. 

Author Response

Dear Reviewer

The Authors reply to:

Comments and Suggestions for Authors

I found this paper very difficult to read.

  • After several readings the methods were still unclear. The authors refer often to the use of the DivGis program to explain the methods which were not well explained  for a non DivGis user which I think should be done for a journal in Agronomy. The methods should be explained without the need to refer to a program

Authors reply: Dear reviewer, in the cases in which it was employed the software Diva Gis, we explained how works the software.  In section 2.2. Spatial delimitation of the regions of genetic richness per maize race, lines 108-110 specify: DivaGis makes a count of the number of collections in each grid and groups the grids into five class intervals by the number of observations present. For purposes of this study, four class intervals were taken with higher number of observations.

Authors reply: In section 2.3. Spatial delimitation of intraspecific maize diversity, lines 131-132 specify: According to the index used, DivaGis counts the number races in each grid and groups the grids in five class intervals in ascendant order by the number of races present

  • In section 2.6 , line 179 the authors state "For  the creation of the index, it was re-categorized ... "  . Replace 'it' with a more descriptive term.

Authors reply: In the text was changed the word re-categorized by the word reclassified, that is a most adequate term. We include this precision in the text.

  • The legend colors in Figure 4 need to be corrected.

Authors reply: This observation was attended, and the shading of Figure 4 was eliminated. We include this precision in the text.

  • I am very concerned about the use of Pearson correlation coefficient to measure associations between the presence of ethnic groups with the number of collections. Pearson's correlation coefficient assumes a simple linear relations between the various measures.  I would like the authors to include a scatterplot matrix  to assure the reader that this statistic is appropriate. I would recommend the scatterplotMatrix function from the 'car' package available in CRAN. 

Authors reply: Dear Reviewer, thanks for this suggestion. To ensure that Pearson's correlation coefficient is appropriate, a Scatterplot matrix was created, The Table 2 was changed by Figure 6. We include this precision in the text.

Authors reply: Figure 6. Scatterplot matrix of Pearson’s correlation coefficient that associates the presence of ethnic groups in the environmental units with the number of collections, number of races, number of principal races, and index of maize uses (grain and plant) (n=7). We include this precision in the text.

Best regards

The Authors